# Tailored-Made Polydopamine Nanoparticles to Induce Ferroptosis in Breast Cancer Cells in Combination with Chemotherapy

**DOI:** 10.3390/ijms22063161

**Published:** 2021-03-19

**Authors:** Celia Nieto, Milena A. Vega, Eva M. Martín del Valle

**Affiliations:** Departamento de Ingeniería Química y Textil, Facultad de Ciencias Químicas, Universidad de Salamanca, 37008 Salamanca, Spain; celianieto@usal.es

**Keywords:** breast cancer, polydopamine nanoparticles, iron, ferroptosis, reactive oxygen species, doxorubicin

## Abstract

Ferroptosis is gaining followers as mechanism of selective killing cancer cells in a non-apoptotic manner, and novel nanosystems capable of inducing this iron-dependent death are being increasingly developed. Among them, polydopamine nanoparticles (PDA NPs) are arousing interest, since they have great capability of chelating iron. In this work, PDA NPs were loaded with Fe^3+^ at different pH values to assess the importance that the pH may have in determining their therapeutic activity and selectivity. In addition, doxorubicin was also loaded to the nanoparticles to achieve a synergist effect. The in vitro assays that were performed with the BT474 and HS5 cell lines showed that, when Fe^3+^ was adsorbed in PDA NPs at pH values close to which Fe(OH)_3_ begins to be formed, these nanoparticles had greater antitumor activity and selectivity despite having chelated a smaller amount of Fe^3+^. Otherwise, it was demonstrated that Fe^3+^ could be released in the late endo/lysosomes thanks to their acidic pH and their Ca^2+^ content, and that when Fe^3+^ was co-transported with doxorubicin, the therapeutic activity of PDA NPs was enhanced. Thus, reported PDA NPs loaded with both Fe^3+^ and doxorubicin may constitute a good approach to target breast tumors.

## 1. Introduction

In recent decades, extensive efforts have been made by the scientific community to find novel, more efficient, effective, and tolerable cancer therapies [1]. In the pursuit of this major goal, nanomedicine is playing a fundamental role in trying to improve the therapeutic activity and selectivity of anticancer agents, focusing on those biological aspects of malignancy that differentiate tumor from normal cells [1,2]. However, most developed nanotherapies are only focused on inducing programmed death of cancer cells and these are able to combat the apoptosis pathway many times, so that multidrug resistance (MDR) occurs more frequently than desired [3]. For this reason, recent studies have suggested that targeting biochemical alterations in cancer cells, in combination or not with the employment of pro-apoptotic agents, may be a feasible and efficacious approach to selectively treat cancer and prevent the apparition of MDR [2,4]. In fact, this combinatorial strategy (metabolic targeting plus conventional drug) is even being used for the therapy of other diseases, such as neurodegenerative disorders [5].

Among the biochemical hallmarks of cancer, one that is sparking great interest for the development of new non-apoptosis-based therapies is iron (Fe) metabolism [3]. Fe is an indispensable element for both malignant and normal cells but, since it is essential for cell proliferation and growth, cancer cells exhibit a stronger dependence on it [6]. In this way, the import, storage and export pathways of Fe, which are tightly regulated in healthy cells (with no excretory route for excess Fe), are perturbed in tumors to maintain increased Fe levels in these tissues [7].

Regarding Fe cellular pathways, it should be noted that this metal binds transferrin (Tf) in the plasma and the resulting complex is taken into cells by transferrin receptor-1 (TfR1)-mediated endocytosis [6,8]. Then, Fe^3+^ is reduced to Fe^2+^ and exported to the cytosol. The majority of cytoplasmic Fe^2+^ is stored within ferritin, but a small pool of Fe^2+^ remains free (LIP) and can be detrimental due to its redox ability, since it can directly catalyze free radical formation via Fenton reaction [9]. When the amount of free Fe^2+^ is excessive and the production of free radicals collapses the cellular defense system [10], a Fe-dependent form of regulated cell death takes places: Ferroptosis [9,11]. 

Thus, in recent years, nanoparticles (NPs) capable to induce a Fe-dependent cell death are being developed for cancer treatment. In the vast majority of cases, they are Fe_3_O_4_ NPs combined or not with chemotherapeutic agents, although gold, silver or silica NPs transporting Fe to induce ferroptosis can also be found in the literature [12]. In any case, all the NPs developed with this intention to date are metallic in nature, with one exception: Polydopamine (PDA) NPs [13,14,15]. 

PDA, which is a nature-inspired polymer, has many interesting properties, such as high affinity to a wide range of metal cations, especially to Fe^3+^, Fe^2+^, and Ca^2+^ [13,16]. This phenomenon was shown in previous works [13], as well as that PDA NP capability to load the Fe^3+^ naturally present in late endo/lysosomes may be related to their intrinsic toxicity to tumor cells [17,18,19]. Based on their ability to chelate Fe^3+^, PDA NPs have begun to be used to transport it and induce ferroptosis in malignant cells [13,14,15]. However, in all studies carried out to date, the pH at which the Fe^3+^-loading was performed was not been given enough importance, and it is essential because determines if Fe^3+^ is free or forming Fe(OH)_3_, and therefore, it can condition nanosystem effectiveness and selectivity.

For that reason, in the current work, Fe^3+^-adsorption in PDA NPs was performed at three different pH values (2.5, 3.1, and 4.5) in order to analyze which of the resulting nanosystems was more effective in selectively inducing ferroptosis in breast cancer cells. Likewise, a mechanism involving Ca^2+^ was proposed that could explain the release of Fe^3+^ from PDA NPs in late endo/lysosomes. Finally, Fe^3+^-burden PDA NPs (PDA NPs@Fe) were also charged with doxorubicin (DOX) to enhance their therapeutic activity. The capacity of this widely used drug to cause mitochondrial Fe accumulation and increase the production of reactive oxygen species (ROS) is well-known [20,21]. For this motive, it was chosen to produce a synergist effect with the chelated Fe^3+^ and cause ROS overproduction [21]. As a result, it was found that when Fe^3+^ was loaded to PDA NPs at the two most acidic pH values (2.5 and 3.1), resulting PDA NPs were more toxic to breast tumor cells but more selective, possibly because most of the loaded Fe^3+^ was in a free form and was not Fe(OH)_3_. Otherwise, it was seen that treatment with all different PDA NPs@Fe increased ROS production in breast cancers cells. Thus, this fact indicated that a process of ferroptosis could be occurring. At last, the antitumor activity of these PDA NPs was enhanced with DOX-loading, and it was demonstrated that a synergist effect took place between the chelated Fe^3+^ and the drug. In this way, when both Fe^3+^ and DOX were charged on PDA NPs (PD NPs@Fe/DOX), the nanosystem obtained reduced breast cancer cell viability in a more remarkable manner. Fortunately, PDA NPs@Fe/DOX did not affect normal cell survival rate in the same extent.

## 2. Results

### 2.1. Preparation and Characterization of PDA NPs

PDA NPs were prepared using the standard procedure of oxidative polymerization of dopamine in a basic aqueous medium containing ethanol (27.7% V/V) and NH_4_OH (2.9% V/V) [13,17,18]. Once synthesized, PDA NPs were characterized by TEM and a size-range histogram was obtained, from which it was determined that NPs had an average size of 111.2 ± 16.2 nm (Figure 1a). Moreover, PDA NP hydrodynamic size was determined by DLS and it was found to be 154.2 ± 28.5 nm (PDI = 0.012) (Figure 1b). Possibly, this last value obtained from PDA NPs in suspension was higher than the average size determined through TEM because of the dehydration that is necessary to prepare TEM samples [22].

### 2.2. pH Effect on the Fe^3+^-Adsorption Capacity of PDA NPs

In prior research in which the Fe^3+^-chelation capacity of the PDA NPs was studied, Fe^3+^-adsorption was performed at pH 4.5, since it was a similar value to that of acidic cell organelles (late endosomes and lysosomes) [13,19]. However, the critical pH for Fe^3+^ to form Fe(OH)_3_ in aqueous solutions is 2.7 [23], so most of Fe^3+^ would be as Fe(OH)_3_ at pH 4.5. Thus, to see if carrying out Fe^3+^-adsorption at pH values closer to 2.7 conditioned PDA NP cytotoxicity by guaranteeing the existence of more free Fe^3+^, this metal was also adsorbed at pH 2.5 and 3.1 in the current work.

For this purpose, Fe^3+^-adsorption was performed with FeCl_3_ solutions with an approximate initial Fe^3+^ concentration of 30 mg/L in all cases. As a result, it was observed that when Fe^3+^ was loaded to PDA NPs at pH 4.5, PDA NPs adsorbed 100 mg Fe^3+^/g NPs (equivalent to 13 mg Fe^3+^/L). Nevertheless, when the adsorption process was carried out at pH 2.5 and 3.1, PDA NPs were able to adsorb 16 and 15 mg Fe^3+^/g NP (equivalent to 2.3 and 2.2. mg Fe^3+^/L), respectively (Table 1). At pH 2.5, most of the Fe^3+^ loaded would be as free cation while, at pH 3.1, since it was a value close to 2.7 but higher than it, part of the Fe^3+^ loaded to the PDA NPs would also be as Fe(OH)_3_. Finally, even though Fe^3+^-loading was much more remarkable at pH 4.5 (possibly because the process was allowed to run for longer time), the vast majority of Fe^3+^ would be as Fe(OH)_3_ at this pH value. Thus, taking into account that Fe(OH)_3_ solubility product constant (Ksp) is 37.0 at room temperature, only 3.52 × 10^−5^ mg/L Fe^3+^ would be as free cation. More information regarding this calculation can be consulted in the Appendix A. 

Next, all PDA NPs@Fe were again characterized by TEM and IR spectroscopy to see if changes in their morphology and chemical structure occurred after Fe^3+^-adsorption. Both characterization techniques were chosen because they complement each other well, although other methods, such as small-angle X-ray scattering (SAXS) [24], also allow obtaining relevant information on the structure of nanoparticle systems. 

Regarding TEM characterization, it was noticed that Fe^3+^-loading at pH 2.5 and 3.1 caused PDA NPs@Fe to become slightly spongier in comparison to bare PDA NPs, a change that was even more noticeable at pH 4.5 (Figure 2).

On the other hand, the main bands of the IR spectra were assigned according to previous characterizations of melanins [13]. After Fe^3+^-binding, the principal changes in the 900–1800 cm^−1^ range occurred in the bands at 1280 and 1620 cm^−1^ (Appendix A). These bands may include the C-OH stretching in a phenol ring and the C=O stretching (non-carboxylic acid), respectively, and their relative intensity was higher for the PDA NPs with Fe^3+^ loaded at pH 2.5 (PDA NPs@Fe_2.5_) than for the rest of PDA NPs. In the 2500–3700 cm^−1^ range (Appendix A), the main changes occurred at 2848, 2920, and 2955 cm^−1^. These bands may correspond to the different vibrational modes of aliphatic groups and decreased in intensity for the PDA NPs with Fe^3+^ loaded at pH 4.5 (PDA NPs@Fe_4.5_), while they were broadened, for instance, for the PDA NPs@Fe_2.5_. 

### 2.3. Fe^3+^-Release and Ca^2+^-Loading from/to the Different PDA NPs@Fe

In previous works, it was already described that PDA formed stable complexes through coordination effects with Fe^3+^ cations [25]. However, under acidic environments, such as those inside late endo/lysosomes, the two adjacent hydroxyl groups coordinating with Fe^3+^ on the PDA benzene ring may competitively combine with protons. This fact could stimulate the release of Fe^3+^ from PDA NPs, which, on the contrary, would be considerably lower at physiological pH (7.4) [14,15]. Further, it was also shown that PDA NPs had great affinity for other cations apart from Fe^3+^. In fact, even though Fe^3+^ was one of the cations for which PDA NPs had the highest affinity, its affinity for Ca^2+^, which is also abundant in the endo/lysosomes [25], was even greater [13]. Since previous investigations demonstrated that Ca^2+^ cations could mediate the deprotonation of the PDA catechol group [26], which accounts for Fe^3+^-complexation [13], it was thought that Ca^2+^ existing in the endo/lysosomes may also stimulate the release of the Fe^3+^ charged in PDA NPs along with the acidic pH. 

For this reason, Fe^3+^-release from PDA NPs@Fe was analyzed by using a lysosome-simulator buffer (pH 4.5) [27], which contained Ca^2+^ in a concentration (20 mg/L) similar to that believed in these organelles [25]. All PDA NPs@Fe were suspended in this buffer for 48 h and, after such time, they were isolated to determine the concentration of Fe^3+^ and Ca^2+^ present in the supernatant and, thus, to find out the amount of Fe^3+^ that was released and the Ca^2+^ that was adsorbed. 

As can be seen from the data collected in Table 1, Fe^3+^ was released from all PDA NPs@Fe after 48 h and all of them were able to adsorb Ca^2+^. Regarding Fe^3+^-release, it should be noted that it was greater from PDA NPs@Fe_4.5_ than from NPs@Fe_2.5_ and NPs@Fe_3.1_. This fact could be consequence of the Fe^3+^ small ionic radius (0.64 Å) and ionic charge, which may hinder its release. Conversely, Fe(OH)_3_ molecules, which were expected to be loaded at pH 4.5, are much larger and neutral, and this fact could make their release easier. 

Otherwise, the differences in Ca^2+^-adsorption capacity were not so marked among the different PDA NPs@Fe. In any case, it was shown that this cation was loaded at the same time that Fe^3+^ was released from the NPs, so an ionic displacement may contribute to the release of the loaded Fe^3+^ from PDA NPs in the endo/lysosomes.

### 2.4. Therapeutic Activity of PDA NPs@Fe as Function of the Fe^3+^-Loading pH

As it has been described before, one of the aims of this work was to verify if loading Fe^3+^ at pH values closer to the critical pH at which Fe(OH)_3_ is formed affected to the cytotoxicity of the PDA NPs@Fe. Herein, the viability of BT474 and HS5 cells after treating them with all the PDA NPs@Fe synthesized was evaluated by MTT assays and compared to that obtained after treating them with bare PDA NPs. BT474 cell line was selected because it had been shown that breast cancer cells have an altered Fe homeostasis that enhances their growth, survival, and metastasis [28,29]. For instance, they overexpress ferritin to store more Fe and TfR1 to enhance its uptake [29]. Accurately, the overexpression of this receptor sensitizes cells to ferroptosis by increasing Fe^3+^-loaded nanosystem endocytosis rate [9,15]. In addition, BT474 cells overexpress HER2, and researchers observed that a complex of Tf with a ferroptosis-inducing compound was capable of declining HER2 expression in this cell line [30]. On the other hand, HS5 stromal cells were chosen to compare PDA NPs@Fe toxicity to malignant and normal cells.

Thus, HS5 and BT474 cells were treated with 0.035 mg/mL PDA NPs and PDA NPs@Fe. The mentioned NP concentration was chosen because in a previous work it was demonstrated that it reduced tumor cell viability but without significantly affecting the survival of healthy cells [19]. Likewise, to load PDA NPs with Fe^3+^ at the different pH values, a solution with an initial 30 mg/L Fe^3+^ concentration was prepared because in an anterior study, when a solution with a higher metal concentration was employed, obtained Fe^3+^-loaded PDA NPs turned out to be toxic to normal fibroblasts [13]. 

The results obtained in these first MTT assays have been represented in the Figure 3 and, to facilitate their comparison, the corresponding viability rate values have been included in the Appendix A. In general, it can be said that all PDA NPs@Fe reduced the survival rate of BT474 cells (Figure 3a) in greater order than unloaded PDA NPs. In this manner, PDA NPs@Fe_4.5_ treatment increased the death of breast cancer cells by approximately 10% compared to naked PDA NPs at all times, reducing their viability by almost half (to 52.8 ± 1.3%) after 72 h. This cytotoxicity increase was close to 20% in the case of the treatment with the other two PDA NPs@Fe, so that the viability of BT474 cells was reduced to 43.3 ± 1.4% (PDA NPs@Fe_2.5_) and 45.6 ± 1.5% (PDA NPs@Fe_3.1_) at the end of the assays, while 63.6 ± 1.9% of BT474 cells were alive 72 h after treatment with unloaded PDA NPs.

Regarding HS5 cell viability (Figure 3b), it was not as affected as that of malignant cells after treatment with any of the PDA NPs@Fe. Their survival rate after treating them with unloaded PDA NPs was close to 90%, 85%, and 80% after 24, 48, and 72 h, and these percentages were very similar to those obtained after PDA NPs@Fe_2.5_ treatment. Thereby, in this case, HS5 cell viability was affected by 35–40% less compared to breast cancer cell viability. PDA NPs@Fe_3.1_ and PDA NPs@Fe_4.5_ were less selective and decreased stromal cell survival rate most notably, especially after 48 and 72 h. Thus, compared to BT474 cells, when HS5 cells were treated with PDA NPs@Fe_3.1_, 40–20% more of them survived and, when they were treated with PDA NPs@Fe_4.5_, this percentage range was 20–10%. 

Consequently, the Fe^3+^-loaded PDA NPs with the most remarkable therapeutic activity were the PDA NPs@Fe_2.5,_ which were also the most selective. On the contrary, PDA NPs@Fe_4.5_ were those that showed an antitumor activity more similar to that of bare PDA NPs and reduced the most the survival rate of the normal cells. As already mentioned, practically all the Fe^3+^ loaded to PDA NPs@Fe_4.5_ was Fe(OH)_3_, while PDA NPs@Fe_3.1_ had both free Fe^3+^ and Fe(OH)_3_ and PDA NPs@Fe_2.5_ had only free Fe^3+^. Since free Fe^3+^ is the one that binds to Tf and is reduced in late endo/lysosomes to the Fe^2+^ that can later participate in Fenton chemistry (Equation (1)) [6,7,8,9], it made sense that PDA NPs with the highest charged amount of free cation were the most efficient and selective, even despite having adsorbed less amount of Fe.
(1)$$Fe^{2+}+\;H_{2}O_{2}\;⟷\;Fe^{3+}\;+\;HO^{-}\;+HO^{\cdot }$$

### 2.5. Therapeutic Activity of PDA NPs@Fe/DOX

To enhance the antitumor activity of PDA NPs@Fe, loading them also with DOX was decided. This drug exerts part of its therapeutic activity through the accumulation of intramitochondrial Fe and the production of ROS. For this reason, PDA NPs were also loaded with it in order to achieve a synergist between Fe^3+^ and DOX [20,21], while trying to reduce DOX severe side effects [31]. In this manner, in order to verify if these two phenomena took place, both PDA NPs and PDA NPs@Fe were loaded with three different concentrations of DOX (0.3, 0.6, and 1 µM). Since the LD50 of this drug for the BT474 cell line was found to be 1 µM [32], it was decided to employ a solution of this concentration to absorb DOX in PDA NPs and PDA NPs@Fe, as well as two solutions with a concentration lower than it. Moreover, there are more and more clinical trials in which some therapeutic nanosystem is administered with some free conventional antitumor drug to achieve a peak of drug concentration in the plasma followed by a more sustained release [33,34,35]. For this reason, it was decided both to isolate the PDA NPs/PDA NPs@Fe once the adsorption process was finished to study their efficacy and selectivity, but also to analyze the cytotoxicity of the whole suspensions (PDA NPs@DOX and PDA NPs@Fe/DOX + unloaded DOX), in which DOX concentrations were expected to be therefore 0.3, 0.6, and 1 µM.

In those NPs that were isolated (PDA NPs@DOX^A^), the amount of DOX that was loaded was determined by measuring the concentration of the drug present in the supernatants by UV-Vis. Results obtained were those of the Table 2. Among the different PDA NPs, there were not remarkable differences in terms of DOX adsorption capacity, although it was noticed that PDA NPs@Fe were capable of adsorbing a little less amount of drug than unloaded PDA NPs.

Then, the effect of both PDA NPs@DOX^A^ and PDA NPs@DOX that were kept with unloaded DOX (PDA NPs@DOX^W^) on the viability of BT474 and HS5 cells was firstly analyzed and compared to that of equivalent concentrations of DOX. 

Focusing first in the results obtained after PDA NPs@DOX^W^ treatment of the BT474 cell line (Figure 4a), it could be seen that it was very effective, especially after 72 h, when it decreased their viability by 60–75%. Moreover, it could be also noticed that the employment of this nanosystem helped to achieve a more-sustained drug release. The drawback was PDA NPs@DOX^W^ also reduced the viability of HS5 cells in a similar way to that of BT474 cells after 48 and 72 h (Figure 4c). Nonetheless, these NPs were certainly much less toxic than equivalent concentrations of free DOX, so PDA NPs@DOX^W^ administration could contribute to reduce DOX severe side effects.

On the other hand, when treatment of breast tumor cells was performed with PDA NPs@DOX^A^ (empty bars) (Figure 4b), their survival rate was not so affected, as expected from working with lower DOX concentrations. However, these NPs were also considerably effective, and reduced BT474 viability by 45–65% after 72 h. In addition, when PDA NPs@DOX^A^ were employed for HS5 cell treatment (Figure 4d), it was observed that these NPs were not so toxic to normal cells, whose viability was 20–25% higher than that of cancer cells at all the times analyzed. 

Otherwise, final MTT assays were carried out with the BT474 and HS5 cell lines to determine the antitumor activity and selectivity of the different PDA NPs@Fe loaded also with DOX. Again, both types of cells were treated with isolated and non-isolated PDA NPs@Fe/DOX (PDA NPs@Fe/DOX^A^ and PDA NPs@Fe/DOX^W^), and the results obtained were those of Figure 5.

Regardless of whether BT474 cell treatment was carried out with PDA NPs@Fe/DOX^A^ or PDA NPs@Fe/DOX^W^, differences in their viability rate could be appreciated again as a function of the Fe^3+^-loading pH. PDA NPs@Fe_3.1_/DOX were the ones with the most remarkable antitumor activity, maybe because they had been shown to be capable of adsorbing more DOX than the other nanosystems. The NPs with a second more marked therapeutic activity, despite being the ones that adsorbed least DOX, were PDA NPs@Fe_2.5_/DOX, possibly because the Fe^3+^ loaded to these NPs was the free cation and not Fe(OH)_3_. At last, PDA NPs@Fe_4.5_/DOX, despite having DOX adsorption efficiencies very similar to those of PDA NPs@Fe_3.1_/DOX, reduced BT474 cell viability in a lesser extent. Nevertheless, all PDA NPs@Fe/DOX turned to be very effective. After 72 h of treatment, PDA NPs@Fe/DOX^W^ reduced tumor cell viability to 7–35% (Figure 5a), according to the type of PDA NPs@Fe and in a DOX concentration-dependent manner and, in the case of PDA NPs@Fe/DOX^A^, this percentage range was 20–42% (Figure 5b).

Besides, PDA NPs loaded with both Fe^3+^ and DOX, showed more noticeable therapeutic activity than PDA NPs loaded only with Fe^3+^ or DOX. Thus, PDA NPs@Fe/DOX were able to further reduce the viability of BT474 cells compared to PDA NPs@Fe and PDA NPs@DOX, and this demonstrated that there was a synergist effect between the metal cation and the drug. To facilitate understanding this fact, viability rate values obtained after BT474 treatment with bare PDA NPs, PDA NPs@Fe_2.5_, PDA NPs@DOX_0.6_, and PDA NPs@Fe_2.5_/DOX_0.6_ have been collected in Figure 6. For instance, after 48 h of treatment, when PDA NPs with adsorbed DOX were administered with unloaded DOX (Figure 6a), PDA NPs@Fe_2.5_/DOX_0.6_^W^ reduced BT474 viability by about 10% more than PDA NPs@Fe_2.5_ and 20% more than PDA NPs@DOX_0.6_^W^. Likewise, when breast cancer cells were treated with PDA NPs with only loaded DOX (Figure 6b), PDA NPs@Fe_2.5_/DOX_0.6_^A^ were 10% and 18% more efficient than PDA NPs@Fe_2.5_ and PDA NPs@DOX_0.6_^A^, respectively, in reducing BT474 viability.

Finally, in addition to being effective, all PDA NPs@Fe/DOX also demonstrated certain selectivity, since when HS5 cells were treated with them (Figure 5c,d), their viability was 15–40% greater than that of BT474 cells. The most selective NPs were PDA NPs@Fe_2.5_/DOX, followed by PDA NPs@Fe_3.1_/DOX and PDA NPs@Fe_4.5_/DOX. Precisely, the content in free Fe^3+^ and not in Fe(OH)_3_ also followed this trend, so that results suggested again that the pH value at which Fe^3+^-adsorption was performed may be important to determine Fe^3+^ state and, therefore, PDA NPs@Fe endocytosis mediated by Tf and TfR1, which are overexpressed in malignant phenotypes but not in normal tissues [36].

All the different viability values represented in Figure 4 and Figure 5 have also been included in the Appendix A.

### 2.6. In Vitro ROS Production after Treatment with PDA NPs and PDA NPs@Fe 

To verify that both DOX and PDA NPs@Fe treatment increased ROS production, the generation of radicals was evaluated 48 h after administering them to BT474 cells by FACS, using a ROS-sensitive probe. As shown in Appendix A, a significant increase in intracellular ROS levels was detected when BT474 cells were treated with DOX and, especially, when NPs loaded with Fe^3+^ were administered. Furthermore, the overproduction of ROS followed the same trend as the viability reduction of breast cancer cells, so that PDA NPs@Fe_2.5_ caused a slightly higher ROS increase than the rest of PDA NPs@Fe, and the population of cells CellROX^−^ Sytox^−^ was smaller after this treatment. 

## 3. Discussion

The fundamental work that nanomedicine is carrying out in the search for novel therapies against cancer is indisputable. In recent years, given the frequent appearance of MDR, this research is moving to the development of nanosystems targeting the biochemical aspects that differentiate tumor from normal cells instead of only developing pro-apoptotic nanomedicines [2,37]. One of these biochemical hallmarks of cancer, which is arousing great interest, is Fe homeostasis. Since Fe is essential for cell growth and proliferation, tumor cells are more dependent on this metal and overexpress the proteins that mediate its uptake and cellular internalization [3,6,7,37]. Based on this fact, the number of nanosystems that are being synthesized to induce targeted Fe-dependent death, known as ferroptosis [9,10,11], is growing considerably [12]. 

PDA NPs constitute one of such nanosystems. It is believed that their great Fe affinity could be related to their intrinsic antitumor activity [17,18,19], and some works can be found in the literature in which Fe^3+^ has been loaded to these NPs to improve their therapeutic effect [13,14,15]. However, in these studies, the pH at which Fe^3+^-adsorption was carried out was not given the importance it deserves, since the pH is essential when determining whether Fe^3+^ is a free cation or Fe(OH)_3_.

For this reason, in this work, it was decided to analyze how the loading pH affected to PDA NPs Fe^3+^-adsorption efficiency and cytotoxicity. Thus, Fe^3+^-adsorption was carried out at the late endo/lysosomal pH (4.5), but also at 2.5 and 3.1, since these pH values were closer to the critical pH at which Fe(OH)_3_ starts to be formed. When Fe^3+^-adsorption was performed at pH 4.5, the amount of Fe^3+^ loaded to PDA NPs was greater than at pH 2.5 and 3.1, perhaps because adsorption was allowed to run during less time at more acidic pH values. Similarly, PDA NPs@Fe_4.5_ showed an enhanced Fe^3+^-release, which could be promoted by the acidic lysosomal pH and the Ca^2+^ existing in these organelles. Nevertheless, despite exhibiting a larger amount of Fe^3+^ loaded and released, PDA NPs@Fe_4.5_ reduced the viability rate of breast cancer cells in a lesser extent than the other PDA NPs@Fe. The explanation could lie in the fact that these NPs practically entirely loaded Fe(OH)_3_, while PDA NPs@Fe_3.1_ would have loaded both free Fe^3+^ and Fe(OH)_3_, and PDA NPs@Fe_2.5_ would only have loaded free Fe^3+^. In this way, although the efficiency of Fe^3+^-adsorption at pH 2.5 and 3.1 was smaller, since the amount of free Fe^3+^ loaded to PDA NPs was greater, so was their therapeutic activity, possibly because their Tf and TfR1-mediated endocytosis may be enhanced [14] (Figure 7a). In addition, since Tf and TfR1 are overexpressed in tumor cells but not under normal physiological conditions, this explanation would also help to understand why PDA NPs@Fe_2.5_ and PDA NPs@Fe_3.1_ were less toxic to stromal cells than PDA NPs@Fe_4.5_.

Otherwise, with the aim of enhancing PDA NPs@Fe antitumor activity, loading them with different concentrations of DOX, while trying to reduce the severe adverse effects of this drug, was determined. Following assays were performed in two different manners. On one hand, PDA NPs@DOX and PDA NPs@Fe/DOX were isolated to study their cytotoxicity and, on the other hand, they were administered as combination therapy with unbound DOX. In both cases, PDA NPs@DOX showed more sustained DOX release and less toxicity than equivalent concentrations of the parent drug. Their antitumor activity increased as function of DOX concentration and, as expected, combination therapy was more effective, but also less selective, than treatment with isolated PDA NPs@DOX^A^. This phenomenon also occurred with PDA NPs@Fe/DOX, which exhibited a more remarkable antineoplastic activity than the PDA NPs in which only Fe^3+^ or DOX had been adsorbed. Such fact showed that there was a synergist effect between the metal and the drug, which was precisely chosen among the different antitumor agents for causing ROS overproduction, as occurs in the ferroptosis process (Figure 7b). This synergy was evident for all PDA NPs@Fe/DOX, but differences in their cytotoxicity could also be appreciated. In general, the NPs that reduced breast cancer cell viability the most were PDA NPs@Fe_3.1_/DOX, while the most selective were PDA NPs@Fe_2.5_/DOX. The first were the PDA NPs that adsorbed the greatest amount of DOX and, PDA NPs@Fe_2.5_/DOX were the particles with less loaded drug. This, along with the fact that these PDA NPs had more free Fe^3+^ loaded than the PDA NPs@Fe_4.5_/DOX, would help to understand their cytotoxic behavior. Finally, PDA NPs@Fe_4.5_/DOX, despite having more DOX burdened than PDA NPs@Fe_2.5_/DOX and more Fe^3+^ loaded than both PDA NPs@Fe_2.5_/DOX and PDA NPs@Fe_3.1_/DOX, were not the most effective, but were the least selective. Such phenomenon revealed once again that the pH at which Fe^3+^-adsorption was performed may condition the efficacy and side toxicity of the resulting PDA nanosystems, which could be potentially chosen or alternatively administered depending on the objectives pursued (more efficacy vs. fewer side effects) and the response of the patients.

## 4. Materials and Methods

### 4.1. Synthesis and Characterization of PDA NPs

PDA NPs were synthesized by the oxidation and self-polymerization of dopamine in a basic aqueous medium. Briefly, deionized H_2_O (H_2_O(d), 90 mL) was mixed with pure ethanol (40 mL) and NH_4_OH (25–28%, 4.2 mL), and the resulting mixture was kept under vigorous stirring for 30 min. Then, dopamine hydrochloride (0.5 g) was dissolved in H_2_O(d) (10 mL) and this solution was added to the previous mixture. The polymerization reaction was allowed to proceed for 24 h and after that time, PDA NPs obtained were isolated by centrifugation [13,17,18]. Next, four centrifugation–redispersion cycles in H_2_O(d) were performed to eliminate any residue, and washed PDA NPs were finally resuspended in H_2_O(d) in a final concentration of 2 mg/mL.

To characterize them, TEM images were taken (Tecnai Spirit Twin, Fei Company, Hillsboro, OR, USA) with a voltage acceleration of 120 kV. PDA NPs were resuspended in H_2_O(d) in a concentration inferior to 0.01% WT, and drops of this dispersion were deposited on copper grids with a collodium membrane. They were allowed to dry for 24 h, TEM images were taken and size-range histograms were obtained after determining the size of at least 300 different PDA NPs (ImageJ software, NIH, Bethesda, MD, USA). In addition, PDA NP hydrodynamic diameter was also determined by DLS (Zetasizer Nano ZS90, Malvern Instruments Inc., Royston, Hertfordshire, UK) on the basis of their intensity-average size distribution. On this occasion, PDA NPs were dispersed in Trizma base solution (pH 10.0), also in a concentration lower than 0.01% WT.

### 4.2. Fe^3+^-Loading to PDA NPs at Different pH Values

To load PDA NPs with Fe^3+^, NPs (1 mL, 2 mg/mL) were mixed with solutions of FeCl_3_ (30 mg/L, 20 mL) that were prepared in buffers of different pH values. Thus, to load Fe^3+^ to PDA NPs at pH 2.5 and 3.1, FeCl_3_ solutions were prepared in citrate/NaOH buffer (0.1 M), whose pH was adjusted to the mentioned values by dropping HCl (0.1 M). Nevertheless, to load Fe^3+^ to PDA NPs at pH 4.5, FeCl_3_ was dissolved in acetate buffer (0.4 M), just as it had been done in previous works [13,19]. In all cases, PDA NPs were kept stirred within the FeCl_3_ solutions at 100 rpm at room temperature but, at pH 2.5 and 3.1, the adsorption process was allowed to run for 3 h, while it was allowed to run overnight at pH 4.5. This time difference was due to the fact that very acidic pH values can alter PDA chemical structure and as a consequence, Fe^3+^-loading at pH 2.5 and 3.1 could not be extended for longer.

To determine the Fe^3+^-adsorption efficiency, PDA NPs@Fe were isolated by centrifugation and the concentration of Fe^3+^ existing in the different supernatants was analyzed by ICP-OES (ULTIMA 2 emission spectrometer, Horiba Jobin Yvon, Unterhaching, Germany). Subsequently, all PDA NPs@Fe were characterized by TEM imaging, preparing the samples in the same way as in the previous point. Moreover, to verify if any modification occurred in the chemical structure of PDA NPs after Fe^3+^-adsorption, IR spectra of PDA NPs@Fe were obtained (SpectrumTwo^TM^, PerkinElmer, Waltham, MA, USA) and compared to bare PDA NP spectrum in the 4000–400 cm^−1^ wavelength range. Samples were prepared as pellets of PDA NPs@Fe in KBr, chosing a weight ratio that did not present saturation in the absorption bands. All spectra were normalized at 1580 cm^−1^ to be able to compare the intensity of the bands.

### 4.3. Fe^3+^-Release and Ca^2+^-Loading from/to the Different PDA NPs@Fe

To later preform in vitro cytotoxicity assays, bare PDA NPs and all PDA NPs@Fe were washed again through five centrifugation–redispersion cycles in PBS (0.01 M, pH 7.4), being finally resuspended in this buffer in a final concentration of 2 mg/mL. Then, PDA NPs@Fe (1 mL) were mixed with CaCl_2_ solutions (20 mg/L, 20 mL) prepared in a lysosomal simulator buffer (0.1 M C_6_H_8_O_7_, 0.2 M Na_2_HPO_4_, pH 4.5) [27], and the resulting suspensions were kept in agitation (100 rpm) for 48 h. After such time, PDA NPs@Fe/Ca were isolated again by centrifugation and the concentrations of Fe^3+^ and Ca^2+^ existing in the supernatants were determined by ICP-OES.

### 4.4. DOX-Loading to PDA NPs and the Different PDA NPs@Fe

Once PDA NPs (with and without loaded Fe^3+^) were resuspended in PBS, DOX-adsorption was carried out. For this, three different volumes of a stock solution of the drug were added to PDA NP suspensions (250 µL, 2 mg/mL) in order to obtain DOX working concentrations of 0.3, 0.6, and 1 µM. The adsorption process was allowed to run overnight in dark conditions, keeping the NPs in agitation (100 rpm). Next day, PDA NPs of some suspensions were isolated by centrifugation and resuspended in PBS (2 mg/mL). However, other suspensions were not centrifuged, and PDA NPs were kept with non-adsorbed DOX to perform two different types of cytotoxicity assays. 

In those PDA NPs that were isolated, the amount of DOX adsorbed was determined by difference, quantifying DOX concentration in the supernatants by UV-Vis spectrophotometry (UV-1800, Shimadzu Corporation, Kioto, Japan). The absorbance of the samples was measured at 480 nm, using the supernatant of the DOX-unloaded PDA NPs as blank. Subsequently, DOX concentration was determined from a previously made calibration curve.

### 4.5. Cell Culture

BT474 and HS5 cell lines (ATCC, Wessel, Germany) were cultured as instructed. They were grown in DMEM, supplemented with FBS (10% V/V) and antibiotics (100 U/mL penicillin and 100 mg/mL streptomycin) in a 95:5 air/CO_2_ humidified atmosphere at 37 °C. 

### 4.6. In Vitro Cytotoxicity

First, the cytotoxicity of PDA NPs and the different PDA NPs@Fe was compared using BT474 and HS5 cells by the MTT assay. BT474 and HS5 cells, once grown, were seeded into 24-well plates (12,000 cells/well) and incubated for 24 h to allow attachment. Then, culture medium in the wells was replaced by fresh supplemented DMEM containing PBS (for the control), PDA NPs, PDA NPs@Fe_2.5_, PDA NPs@Fe_3.1_, or PDA NPs@Fe_4.5_ (0.035 mg/mL), and cells were incubated again for 24, 48 and 72 h. A total of 110 µL MTT solution (5 mg/mL) were added to each well and cell incubation was performed for 1 h. Formazan crystals were dissolved by adding DMSO (500 µL/well), and the optical density value of each well was determined by applying a procedure that was previously set-up to subtract PDA contribution to sample absorbance [17]. This absorbance was measured in a microplate reader (Zetasizer Nano ZS90, Malvern Instruments Inc., Royston, UK). 

Next, the same protocol was followed to assess the cytotoxicity of PDA NPs@DOX and PDA NPs@Fe/DOX, which was compared to that of free DOX (0.3, 0.6 and 1 µM). In all cases, BT474 and HS5 cells were treated with 0.035 mg/mL PDA NPs but, as mentioned before, these assays were performed in two different manners: On one hand, cells were treated with PDA NPs@DOX or PDA NPs@Fe/DOX that were isolated after the DOX adsorption process and, on the other hand, they were treated with the complete suspension, in which PDA NPs@DOX or PDA NPs@Fe/DOX were kept with unloaded DOX. Again, the viability of BT474 and HS5 cells was analyzed each 24 h, for a total of 72 h, following the same steps as before.

### 4.7. In Vitro ROS Detection

ROS production was evaluated 48 h after BT474 treatment with DOX (0.3 µM) and PDA NPs@Fe (0.035 mg/mL) by using CellROX^®^ Deep Reagent (Thermo Fisher, C10491). As described in the manufacturer’s instructions, BT474 cells were harvested and CellROX^®^ (0.5 µM) was added to the suspension (1 mL, 10^6^ cells/mL). Cells were incubated for 45 min at 37 °C and Sytox^®^ (1 µM) was incorporated when 30 min elapsed. Samples were acquired (at least 50,000 events) with a FACSAria^TM^ III cytometer (BD Biosciences, San José, CA, USA) and data obtained were analyzed using CellQuest software (BD Biosciencies). Median values of fluorescence intensity were used to provide a semiquantitative assessment of ROS production [38].

### 4.8. Statistical Analysis

Data concerning MTT assays were analyzed using an unpaired two-tailed Student *t*-test. *p*-values less than 0.05 were considered to be statistically significant. Displayed viability results are the average ± SEM of three replicates per treatment condition that were obtained in two parallel experiments.

## 5. Conclusions

In conclusion, the importance that the pH has when Fe^3+^ is loaded to PDA NPs to induce ferroptosis in tumor cells has become clear throughout this work. This pH value determines Fe^3+^ state and, therefore, the selectivity and therapeutic activity of the resulting Fe^3+^-loaded PDA NPs. Such antitumor activity can be enhanced by co-loading other agents that also increase ROS production in cancer cells, like DOX, and PDA NPs with different concentrations of Fe^3+^ and drugs could be tailored-synthesized and administered depending on the therapeutic need. 

## Figures and Tables

**Figure 1 ijms-22-03161-f001:**
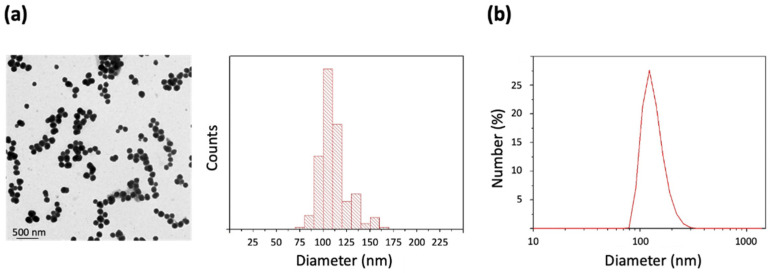
(**a**) TEM image and size-range histogram of the polydopamine nanoparticles (PDA NPs) synthesized; (**b**) DLS number distribution of a suspension of the obtained PDA NPs in Trizma base solution (pH 10.0).

**Figure 2 ijms-22-03161-f002:**
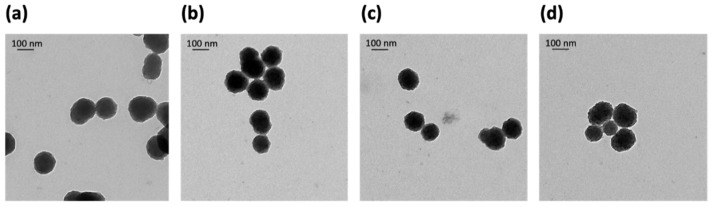
TEM images of (**a**) bare PDA NPs (114.9 ± 17.6 nm); (**b**) PDA NPs@Fe_2.5_ (112.2 ± 19.1 nm), (**c**) PDA NPs@Fe_3.1_ (106.2 ± 17.5 nm) and (**d**) PDA NPs@Fe_4.5_ (111.8 ± 51.1 nm). As shown, Fe^3+^-loading made PDA NPs spongier, especially when the metal adsorption was performed at pH 4.5.

**Figure 3 ijms-22-03161-f003:**
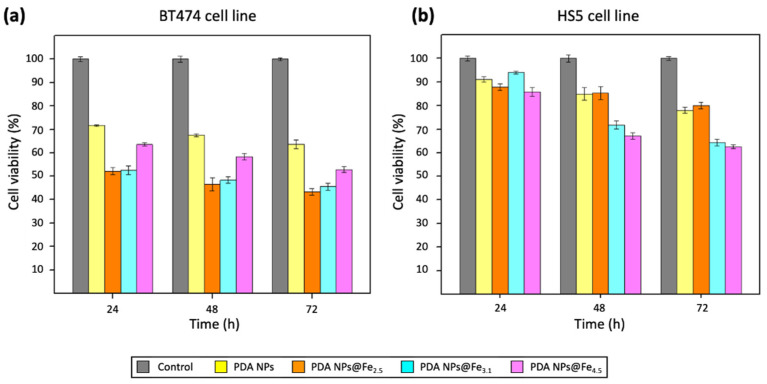
Viability results obtained after the performance of MTT assays with the (**a**) BT474 and (**b**) HS5 cell lines, which were treated with 0.035 mg/mL PDA NPs, PDA NPs@Fe_2.5_, PDA NPs@Fe_3.1_ and PDA NPs@Fe_4.5_.

**Figure 4 ijms-22-03161-f004:**
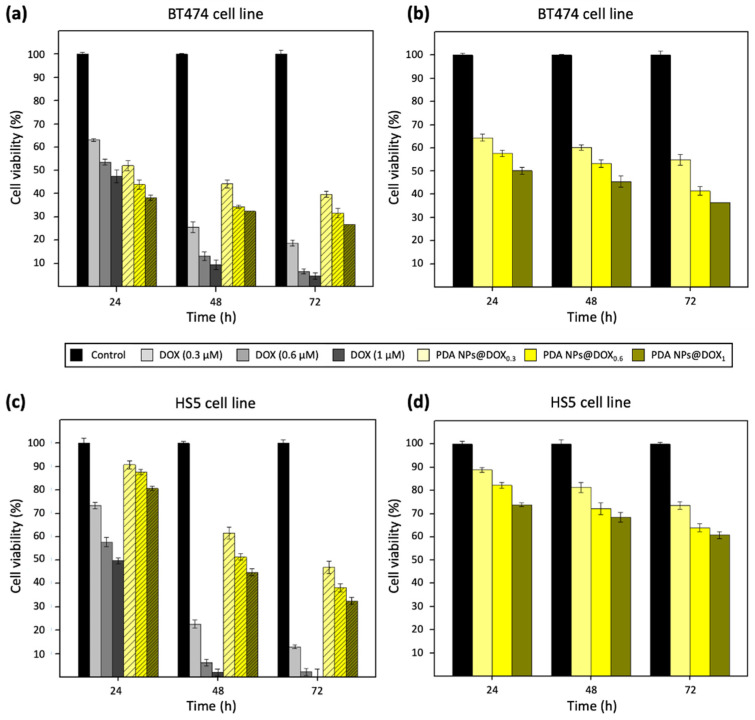
Viability results obtained after the performance of MTT assays with the (**a**,**b**) BT474 and (**c**,**d**) HS5 cell lines, which were treated with DOX (0.3–1 µM) and PDA NPs@DOX_0.3_, PDA NPs@DOX_0.6_ and PDA NPs@DOX_1_ (0.035 mg/mL). Bars with the line pattern represent the viability results that were obtained after using PDA NPs@DOX^W^ (**a**–**c**), while the results obtained with PDA NPs@DOX^A^ were those of the empty bars (**b**–**d**).

**Figure 5 ijms-22-03161-f005:**
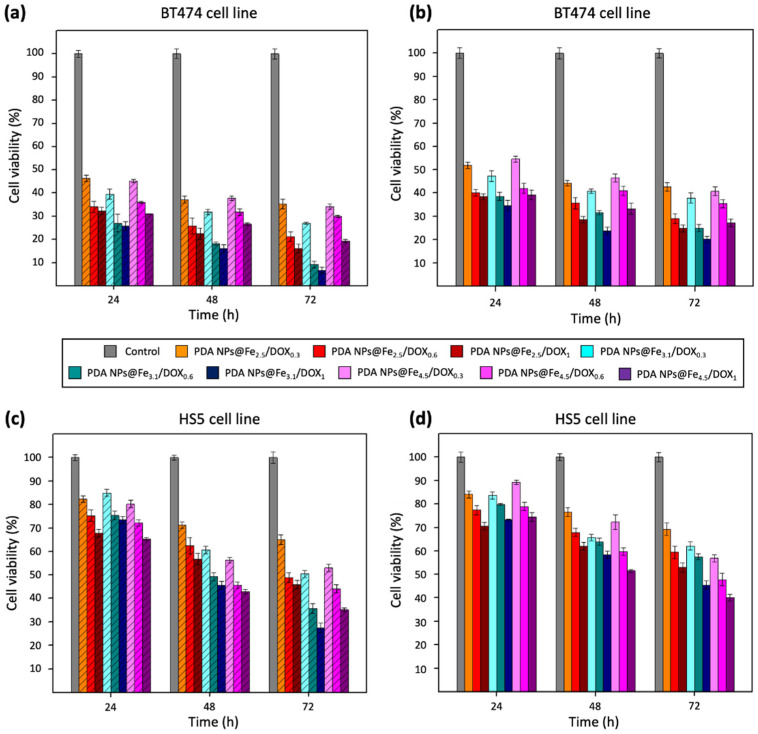
Viability results obtained after the performance of MTT assays with the (**a**,**b**) BT474 and (**c**,**d**) HS5 cell lines, which were treated with all the different PDA NPs@Fe/DOX (0.035 mg/mL). Results of the graphs (**a**,**c**) were obtained after using PDA NPs@Fe/DOX^W^, while PDA NPs@Fe/DOX^A^ were employed to obtain graphs (**b**,**d**).

**Figure 6 ijms-22-03161-f006:**
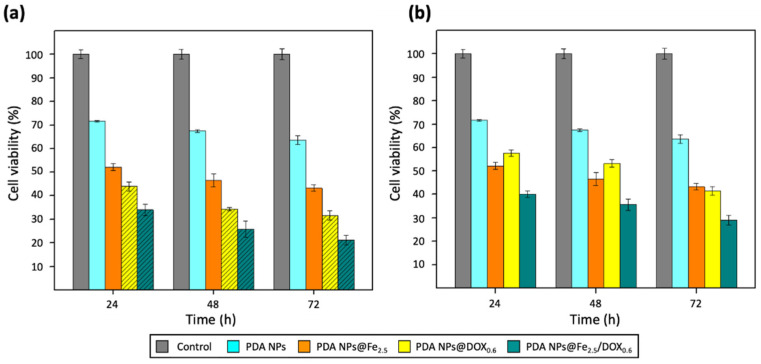
BT474 cell viability rates obtained after the performance of different MTT assays that showed that there was a synergist effect between the Fe^3+^ and doxorubicin (DOX) charged on PDA NPs. Again, bars with the line pattern represent the viability values obtained after BT474 cell treatment with PDA NPs@DOX^W^ (**a**), while the results obtained with the PDA NPs@DOX^A^ are those of the empty bars (**b**).

**Figure 7 ijms-22-03161-f007:**
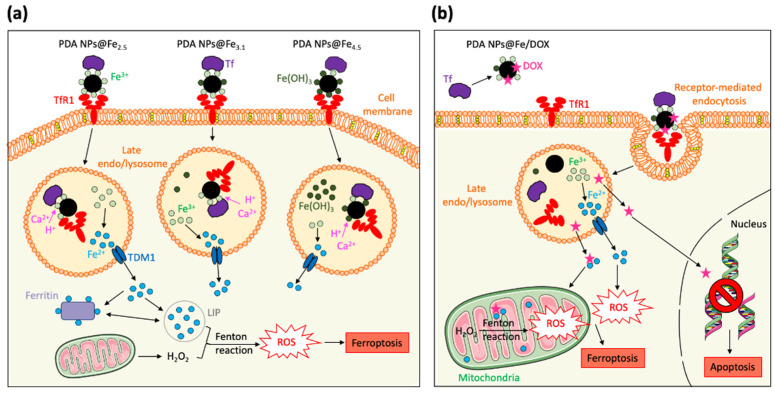
Graphical representation of (**a**) how the pH at which Fe^3+^ was adsorbed may condition the cytotoxicity of the PDA NPs@Fe and of (**b**) the synergist effect that could take place between Fe^3+^ and DOX when both of them were loaded to PDA NPs.

**Table 1 ijms-22-03161-t001:** Amount of the Fe^3+^ released and the Ca^2+^ adsorbed in the different PDA NPs@Fe suspended in lysosomal buffer (pH 4.5) containing Ca^2+^ (20 mg/L). The percentage values regarding Fe^3+^-release were calculated taking into account the amount of this cation that was loaded initially, and those regarding Ca^2+^-adsorption were calculated taking into account Ca^2+^ initial concentration in the buffer.

pH	Fe^3+^ Loaded (q)	Fe^3+^ Released (q)	Fe^3+^ Released (%)	Ca^2+^ Loaded (q)	Ca^2+^ Loaded (%)
2.5	16 mg/g	6 mg/g	38	21 mg/g	13
3.1	15 mg/g	4 mg/g	30	20 mg/g	13
4.5	100 mg/g	75 mg/g	75	16 mg/g	10

**Table 2 ijms-22-03161-t002:** Amount of DOX adsorbed to PDA NPs and PDA NPs@Fe as a function of the DOX initial concentration employed in the adsorption process. Results are expressed as ng DOX/mg PDA NPs.

Initial [DOX]	PDA NPs	PDA NPs@Fe_2.5_	PDA NPs@Fe_3.1_	PDA NPs@Fe_4.5_
0.3 µM	21.4 ng/mg	16.8 ng/mg	19.0 ng/mg	18.8 ng/mg
0.6 µM	42.1 ng/mg	33.2 ng/mg	37.3 ng/mg	35.7 ng/mg
1 µM	57.9 ng/mg	51.7 ng/mg	55.8 ng/mg	54.6 ng/mg

## Data Availability

The data that support the findings of this study are available from the corresponding authors upon reasonable request.

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
