# Peer review of "Tailored-Made Polydopamine Nanoparticles to Induce Ferroptosis in Breast Cancer Cells in Combination with Chemotherapy"

_ijms, 2021, doi:10.3390/ijms22063161_

Round 1

Reviewer 1 Report

The authors designed polydopamine nanoparticles (PDA NP) capable of encapsulating iron and doxorubicin for the purposes of anticancer drug delivery. In vitro tests were performed with the BT474 and HS5 cell lines demonstrating their enhanced therapeutic activity. The reported PDA NPs are important in the context of targeted cancer therapy and dual and multi drug delivery systems.

Before publication I found several important points for a minor revision:

  1. Fig.2 shows a transmission electron microscopy image of the obtained PDA nanoparticles. I did not see a key difference between the loaded and blank nanoparticles. There are methods such as small angle X-ray scattering (SAXS) that are commonly used for the structural characterization of drug delivery nanoparticles. In case the authors have SAXS data it will be beneficial to include them in the paper. Otherwise, they can mention in the discussion or introduction that SAXS and TEM are complementary methods commonly used together for structural characterization of nanoparticles. Reference: Dong, Y.-D.; Boyd, B.J. Applications of X-ray scattering in pharmaceutical science. Int. J. Pharm. 2011, 417, 101–111
  2. The discussion could more clearly denote the dual or multi drug nature of the proposed PDA nanoparticles. In this relation the addition of recent references on the topic will expand the potential of the article to attract a broader readership and interest in other types of cancer therapies : (1) Combination therapies induce cancer cell death through the integrated stress response and disturbed pyrimidine metabolism, EMBO Mol Med (2021), e12461https://doi.org/10.15252/emmm.202012461, (2) Dual and multi-drug delivery nanoparticles towards neuronal survival and synaptic repair, Neural Regeneration Research, 2017, 12, 886-889, DOI: 10.4103/1673-5374.208546.

Author Response

Fig.2 shows a transmission electron microscopy image of the obtained PDA nanoparticles. I did not see a key difference between the loaded and blank nanoparticles. There are methods such as small angle X-ray scattering (SAXS) that are commonly used for the structural characterization of drug delivery nanoparticles. In case the authors have SAXS data it will be beneficial to include them in the paper. Otherwise, they can mention in the discussion or introduction that SAXS and TEM are complementary methods commonly used together for structural characterization of nanoparticles. Reference: Dong, Y.-D.; Boyd, B.J. Applications of X-ray scattering in pharmaceutical science. Int. J. Pharm. 2011, 417, 101–111.

Thanks for your suggestion. The main objective pursued when characterizing polydopamine nanoparticles (PDA NPs) by TEM was analyzing their size and morphology. From the images that were obtained, it could be observed that PDA NP surface seemed more spongy when Fe3+was adsorbed to them at pH 4.5, a phenomenon that had been reported in a previous work [1], where it had been found that Fe-adsorption, in comparison to the loading of other metals, made PDA NPs fluffier.

Otherwise, the method that was employed to identify the main PDA functional groups in both samples of bare PDA NPs and PDA NPs loaded with Fe3+ at the different pH values was FT-IR. On this occasion, no very significant changes could be appreciated in PDA characteristic bands after Fe3+-adsorption regardless of the pH value.

As regards SAXS technique, we unfortunately cannot have data because we do not have the necessary equipment. However, we have found in the literature that this characterization method is normally employed to analyze nanoparticulate systems that are modified during the synthesis process and, therefore, which are integrated by more than one component (for example, graphene oxide, silver or silica plus polydopamine). Since the PDA NPs that we prepared were not modified during the synthesis process, just loaded with Fe3+, we believe that characterizing them by TEM plus FT-IR may be enough, given that these methods complement each other.

In any case, we consider that SAXS is one of the most versatile methods for the structural characterization of drug delivery systems, and we will be happy to mention this fact in the manuscript. Thus, we have included a statement mentioning this fact in the lines 135-137, and we have also incorporated the reference that you have suggested to us (reference 24).

[1] Vega et al. Cytotoxicity of paramagnetic cations-loaded polydopamine nanoparticles. Colloid. Surf. B Biointerfaces 2018, 167, 284-90.

The discussion could more clearly denote the dual or multi drug nature of the proposed PDA nanoparticles. In this relation the addition of recent references on the topic will expand the potential of the article to attract a broader readership and interest in other types of cancer therapies : (1) Combination therapies induce cancer cell death through the integrated stress response and disturbed pyrimidine metabolism, EMBO Mol. Med. (2021), e12461https://doi.org/10.15252/emmm.202012461, (2) Dual and multi-drug delivery nanoparticles towards neuronal survival and synaptic repair, Neural Regeneration Research, 2017, 12, 886-889, DOI: 10.4103/1673-5374.208546.

Thank you again for your comments. We will be glad to include this pair of references in the introduction of the manuscript (references 4 and 5).

Reviewer 2 Report

In this manuscript, the authors prepared a polydopamine nanoparticle to carry  the Fe3+ for the ferroptosis. The article is very interesting. However, some suggestions are as following:

  1. The biological results (cytotoxic results) is suggested to statistically  calculate (for example, t test).
  2. The idea is similar to the article: 

    Fe2+/Fe3+ Ions Chelated with Ultrasmall Polydopamine Nanoparticles Induce Ferroptosis for Cancer Therapy.

    Please compare and discuss the article with the manuscript.

Author Response

The biological results (cytotoxic results) are suggested to be statistically calculated (for example, t test).

Thank you for your comment. We have now included the statistical analysis that we have performed in the materials and methods section (point 4.10 of the manuscript).

The idea is similar to that of the article: Fe2+/Fe3+ Ions Chelated with Ultrasmall Polydopamine Nanoparticles Induce Ferroptosis for Cancer Therapy. Please compare and discuss the article with the manuscript.

Thank you for your suggestion.

In their work, Chen and coworkers synthesized polydopamine nanoparticles (PDA NPs) modified with poly(ethylene-glycol) (PEG) and chelated them with Fe2+ or Fe3+ to induce cancer cell death by ferroptosis. To load PDA NPs with both Fe ions, they prepared solutions of FeCl2 and FeCl3 in Millipore water, and they studied the cytotoxicity of the resulting NPs to human primary glioblastoma and breast cancer cells, which were treated with 0.025 – 0.5 mg/mL of PEG-modified PDA NPs loaded or not with Fe2+/3+ ions. 24 hours after treatment, these authors found that bare PEG-PDA NPs had no obvious cytotoxic effects, and that those chelated with Fe2+ and Fe3+ reduced cancer cell viability to 85-90% when they were administered in a 0.035 mg/mL concentration, which was the same that we employed in our study.

In this manner, the first difference that exists between the work carried out by Chen et al. and ours is that we did not modify PDA NPs, but we directly loaded them with Fe3+ to induce ferroptosis in breast cancer cells. The second difference, and perhaps the most important, is that we studied how the pH at which Fe3+-adsorption was performed determined its state and, consequently, the cytotoxicity of the chelated PDA NPs. Thus, based on our results, if it is possible that if Chen and colleagues had adsorbed Fe3+ at a pH value close to 2.7, they would have obtained more cytotoxic nanoparticles. Finally, it is also important to point out that, when carrying out MTT assays with PDA, the contribution of this polymer to the absorbance of the samples must be taken into account to avoid overestimating cell viability [1]. Nevertheless, Chen et al. did not make this consideration and this, together with having loading Fe2+/Fe3+ at a neutral-basic pH could be responsible for the fact that their PDA NPs reduced the viability of tumor cells by 10-15% after 24 hours, while we found that the same concentration of PDA NPs@Fe3+ reduced it by 50-60% the next day. Furthermore, another notable difference between the work of these authors and ours is that we proposed simultaneously loading PDA NPs with Fe3+and doxorubicin (DOX) to enhance their therapeutic activity, which therefore could be modulated as a function of the Fe3+-adsorption pH and the drug concentration used. In contrast, Chen et al. did not employ any antitumor drug and did not propose a combination therapy, which could help prevent the apparition of resistances.

For all these reasons, we believe that our publication differs considerably form that of Chen et al and we emphasized in the introduction and discussion sections the importance of the pH when it came to adsorbing Fe3+ to PDA NPs (lines 68-71, 375-377 and 423-427).

[1] Nieto et al. Polydopamine nanoparticles kill cancer cells. RSC Adv. 2018, 8, 36201.